# Knowledge, attitude, and practice related to the COVID-19 pandemic among undergraduate medical students in Indonesia: A nationwide cross-sectional study

**Imam Adli**[1], **Indah Suci Widyahening**[2]*, **Gilbert Lazarus**[1], **Jason Phowira**[1], **Lyanna Azzahra Baihaqi**[1], **Bagas Ariffandi**[1], **Azis Muhammad Putera**[1], **David Nugraha**[3], **Nico Gamalliel**[1], **Ardi Findyartini**[4,5]

**1** Faculty of Medicine, Universitas Indonesia, Jakarta, Indonesia, **2** Department of Community Medicine, Faculty of Medicine, Universitas Indonesia, Jakarta, Indonesia, **3** Faculty of Medicine, Universitas Airlangga, Surabaya, Indonesia, **4** Department of Medical Education, Faculty of Medicine, Universitas Indonesia, Jakarta, Indonesia, **5** Medical Education Center, Indonesia Medical Education and Research Institute, Faculty of Medicine, Universitas Indonesia, Jakarta, Indonesia

* indah_widyahening@ui.ac.id

**Data Availability Statement:** All relevant data are within the paper and its Supporting Information files.

## Abstract

### Introduction

The potential role of medical students in raising awareness during public health emergencies has been acknowledged. To further explore their potentials as public educators and role models for the communities during the coronavirus disease 2019 (COVID-19) pandemic, this study aims to assess the knowledge, attitude, and practice of these students toward COVID-19.

### Methods

An online cross-sectional survey was conducted among undergraduate medical students in Indonesia. Socio-demographics characteristics, social interaction history, information-seeking behavior, as well as knowledge, attitude, and practice toward COVID-19 were collected through a self-reported questionnaire. A p-value of <0.05 indicated statistical significance.

### Results

Out of 4870 respondents, 64.9% had positive attitude and 51.5% had positive practice toward COVID-19, while only 29.8% had adequate knowledge. Knowledge was slightly positively correlated with attitude and practice ($\rho = 0.074$ and $\rho = 0.054$, respectively; both p<0.001), while attitude was weakly correlated with practice ($\rho = 0.234$, p<0.001). Several factors including age, sex, place of residence, institution type, academic level, family income, history of chronic illness, prior volunteering experience, and perceptual awareness on COVID-19 were significantly associated with either knowledge, attitude, and/or practice toward COVID-19. Furthermore, health institution's and the government's press releases, as well as health expert opinions were deemed as the most reliable sources of COVID-19-related information–yet trivially none of these sources were associated with knowledge, attitude, and practice in the study population.

**Funding:** AF received funding from the Indonesian COVID-19 Research and Innovation Consortium (grant number: 132/FI/P-KCOVID-19.2B3/IX/2020) organized by the Indonesian Ministry of Research and Technology/National Research and Innovation Agency. The funder had no role in the study conception, data collection and analysis, decision to publish, or preparation of the manuscript.

**Competing interests:** The authors have declared that no competing interests exist.

## Conclusion

Many undergraduate medical students in Indonesia had positive attitude and practice against COVID-19, yet only a few had adequate knowledge. This warrants further interventions to keep them updated with COVID-19 evidence to maximize their potentials in raising public awareness on COVID-19.

## Introduction

Since the declaration of the coronavirus disease 2019 (COVID-19) as a pandemic in early 2020 [1], global communities have strived to implement various strategies in mitigating the devastating disease burden caused by the virus. In Indonesia, the government has prompted several unprecedented measures to control the spread of the disease [2], including implementing large-scale social distancing, increasing the capacity of COVID-19 diagnostic tests, and launching a national research consortium to accelerate innovations to combat the disease [3, 4]. Despite these, the COVID-19 disease spread remains concerning as the number of cases and deaths resulting from the disease has perpetually surged–rendering Indonesia as the hardest-hit country in the region [2]. This may partly be attributable to the fact that COVID-19 literacy among Indonesian population is still poor [5, 6]. Although a previous study revealed that Indonesian population has had positive behaviors toward COVID-19 prevention, some of the essential preventive measures were still lacking, especially in terms of social distancing, self-isolation, maintenance of healthy lifestyle, and health-seeking behavior [5]. This indicated that further strategies to increase the public's awareness and preventive behaviors on COVID-19 are imperative, in which medical students may provide unyielding support by taking part in raising awareness on COVID-19.

The role of medical students in raising awareness during public health emergencies have long been established. A systematic review by Martin et al revealed that medical students have been involved in public health campaigns during previous viral outbreaks, including human immunodeficiency virus, influenza, severe acute respiratory syndrome, and Ebola [7]. Being perceived as having a higher level of health literacy, medical students may serve as role models for the public to adopt COVID-19 preventive health behaviors. During the COVID-19 pandemic, Indonesian medical students have also taken part in disseminating health information to the public, mainly through the use of social media and news outlets [8]. However, as the disease burden of COVID-19 in Indonesia persists despite rigorous efforts, comprehensive assessments of medical students' knowledge, attitude, and practice are imperative to further enhance their potentials in educating the public. Specifically, this may provide crucial information for stakeholders to identify field gaps and devise strategies to further encourage communities to follow health standards. Furthermore, this information may also be utilized by medical institutions to improve the medical curricula to prepare for future outbreaks. Therefore, this study aims to assess the knowledge, attitude, and practice of Indonesian undergraduate medical students toward the COVID-19 disease.

## Methods

### Study design and population

A cross-sectional survey among undergraduate medical students in Indonesia was conducted from 13 July to 11 October 2020. Participants were recruited using a snowball sampling

technique and all data were collected via an online self-reported questionnaire using Google Forms (http://forms.google.com/) as the data collection period coincided with implementation of the COVID-19 lockdown policy in Indonesia. The questionnaire was filled anonymously, voluntarily, and with written consent given by all respondents. To avoid duplicate responses, all participants were required to log into their email accounts, but any personal contact information were not recorded to protect the anonymity of the respondents. During the data collection period, the questionnaire was spread using social media platforms every three days to increase the number of respondents. All procedures conducted in this study have been approved by the Health Research Ethics Committee of the Faculty of Medicine Universitas Indonesia and Cipto Mangunkusumo Hospital (ethical clearance number: 758/UN2.F1/ETIK/ PPM.00.02/2020).

## Measurement tools

The questionnaire was developed through literature searches of previously validated questionnaires [9–12], which was then translated to Bahasa Indonesia through backward and forward translations by three of the authors (GL, IA and AMP), and modified to match the context of the study. A preliminary survey was then conducted on 30 participants to enhance the comprehension and the validity of the questionnaire. Further details on the calculation of minimum sample size and the development of the questionnaire have been previously discussed elsewhere [13].

The questionnaire encompassed four primary sections including: (1) socio-demographics characteristics, (2) social interaction history, (3) levels of trust in COVID-19 health information sources, and (4) knowledge, attitude, and practice. Socio-demographics characteristics consisted of age, sex, place of residence, institution type, academic level, living circumstances, marital status, family income, and history of chronic diseases. Living circumstances comprised the number of people living in the household and type of household (e.g., family, non-family, or alone), while family income was classified as low, lower-middle, upper-middle, and high-income class for monthly family income of ≤ IDR 1,500,000, IDR 1,500,001–2,500,000, IDR 2,500,001–3,500,000, and >IDR 3,500,000, respectively [14]. On the other hand, the social interaction history section aimed to assess the volunteering experience of respondents in health and non-health sectors, their own and family's COVID-19 disease history, and their history of physical contacts with COVID-19 patients. In addition, we also assessed the medical students' general perception on the reliability of various health information sources, including televisions, newspapers, online news, social media, official statements from the government and health institutions, and expert opinions.

The final section aimed to investigate the participants' knowledge, attitude, and practice toward the COVID-19 disease. Knowledge was assessed using a 10-items questionnaire encompassing various areas of COVID-19 disease, including pathogenesis, clinical presentation, diagnosis, treatment, and prevention. Each correct answer accounts for one point with a maximum score of 10 points. In contrast, assessment of attitude and practice consisted of 12 questions each using a five-point Likert scale. Higher knowledge, attitudinal, and behavioral scores indicated favorable perceptions. Lastly, the reliability of the questionnaire was appraised using Cronbach's alpha, with a coefficient for knowledge, attitude, and practice of 0.655, 0.726, 0.807, respectively, indicating satisfactory internal reliability [15].

## Statistical analysis

Submitted responses were collected in and managed with the MS Excel® for Office 365 MSO ver. 2002 (Microsoft Corporation, Redmond, WA, 2018). Subsequently, data were analyzed

using SPSS 24.0 (SPSS Inc., Chicago, IL) and visualized using R ver. 4.0.3 (R Foundation for Statistical Computing, Vienna, Austria) [16]. Categorical data were presented as frequency and percentages, while continuous data as means or medians along with the appropriate measure of dispersion according to the normality of data distribution as tested with Kolmogorov-Smirnov tests.

Outcomes on the knowledge, attitude, and practice were dichotomized according to the Bloom's cut-off (≥80%) [17], where a sum knowledge score of ≥8 indicated adequacy, and a score of ≥48 indicated positive attitude and practice, respectively. Potential factors associated with the dependent variables were first analyzed using univariate logistic regression. Any factors associated with each outcome at p≤0.20 were deemed eligible for inclusion in the multivariate analysis. In addition, correlation between trust in health information sources and knowledge, attitude, and practice, as well as intercorrelations among the dependent variables were investigated with Spearman's rho (ρ). A p-value of <0.05 denoted statistical significance.

## Results

Out of 4870 participants with a median age of 20 years (interquartile range [IQR]: 19–21), 3399 (69.8%) were female and 3925 (80.6%) were pre-clinical students. Most participants resided in Java (86.9%), followed by Eastern Indonesia (6.0%), Sumatra (4.9%), and Central Indonesia (2.1%). Further details on the characteristics of the included participants can be seen on **S1 Table**.

### Knowledge, attitude, and practice toward COVID-19

We discovered that about 64.9% and 51.5% respondents yielded positive attitude and practice toward COVID-19, while only 29.8% yielded adequate knowledge score. Most of the participants were knowledgeable in the prevention (97.1%), respiratory transmission (96.4%), clinical and radiological findings (83.9% and 80.8%), and therapeutic aspects of COVID-19 (83.3%). However, low numbers of correct answers were found in questions related to the laboratory findings (31.8%), zoonotic transmission route of SARS-CoV-2 (31.8%), and the differences in clinical manifestations between COVID-19 and common cold (57.1%; **S2 Table**). Furthermore, only 16.1% of the participants correctly answered the incubation period of SARS-CoV-2. In addition, a considerable number of participants believed that the communities' awareness on COVID-19 were still lacking (37.0%) and disagreed that COVID-19 patients could be treated at home (22.5%). Item-specific responses of the attitude and practice of Indonesian medical students toward COVID-19 can be seen on **S3** and **S4** **Tables**, respectively.

In this study, we found that the correlation between knowledge with attitude and practice was negligible (ρ = 0.074 and ρ = 0.054, respectively; both p<0.001). Attitude was also weakly correlated with practice (ρ = 0.234, p<0.001; **Fig 1**). Our findings indicated that female (OR 1.22 [95% CI: 1.06–1.40], p = 0.006) and students with an older age were more likely to demonstrate a higher level of knowledge (OR 1.06 [95% CI: 1.00–1.12], p = 0.034; **Table 1**). Furthermore, students attending public institutions had a higher knowledge score than those attending private institutions (OR 1.32 [95% CI: 1.15–1.50], p<0.001). We also found that clinical-year students (OR 1.66 [95% CI: 1.35–2.04], p<0.001) and students with prior volunteering experience, either in health sectors (OR 1.26 [95% CI: 1.07–1.49], p = 0.007) or non-health sectors (OR 1.33 [95% CI: 1.15–1.52], p<0.001), were associated with higher knowledge scores. Students with a history of chronic illness also had a higher knowledge score (OR 1.33 [95% CI: 1.04–1.70], p = 0.022). In contrast, students living in Sumatra and Eastern Indonesia had significantly lower knowledge scores relative to those living in Java (OR 0.73 [95% CI: 0.54–0.99], p = 0.044; and OR 0.73 [95% CI: 0.55–0.97], p = 0.028, respectively).

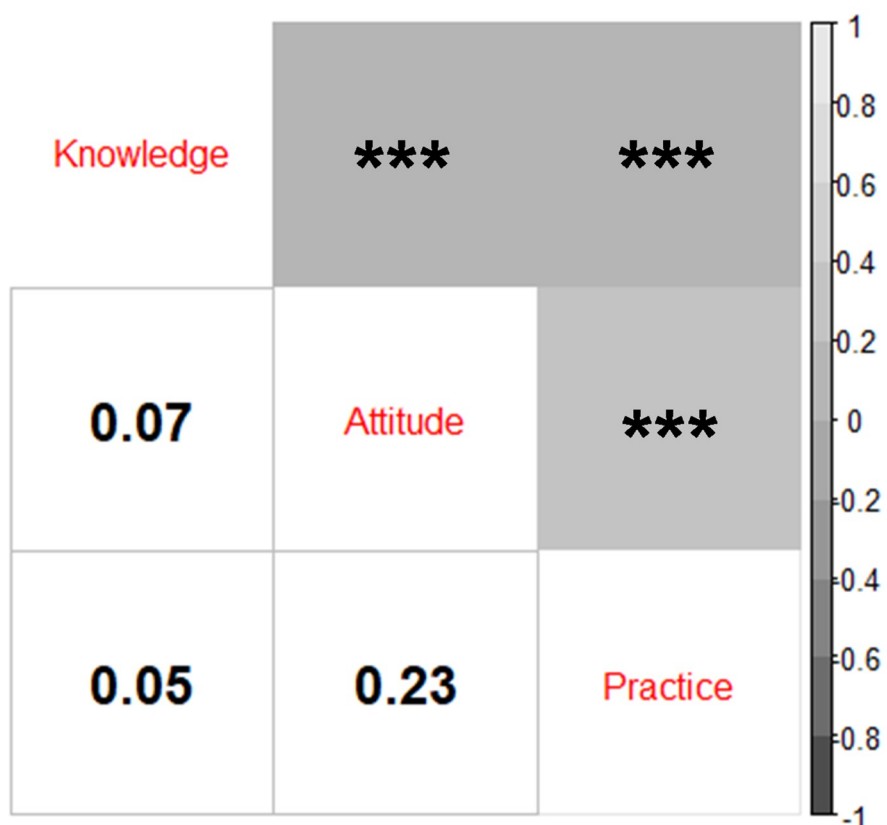

**Fig 1. Correlation between knowledge, attitude, and practice toward COVID-19 in the study population.**
***p<0.001.

Consistent with our findings in the knowledge section, students with prior volunteering experience in either health or non-health sectors yielded higher attitude and practice scores (health sectors: OR 1.19 [95% CI: 1.00–1.41], p = 0.045 and OR 1.25 [95% CI: 1.07–1.47], p = 0.006; non-health sectors: OR 1.25 [95% CI: 1.10–1.41], p = 0.001 and OR 1.16 [95% CI: 1.03–1.31], p = 0.019; **Tables 2** and **3**, respectively). Furthermore, female students also had better practice (OR 1.46 [95% CI: 1.28–1.65], p<0.001). However, contrary to our findings in the knowledge section, we discovered that students in public universities had a relatively lower practice score compared students in private universities (OR 0.71 [95% CI: 0.63–0.80], p<0.001). Students from lower-middle income class also had lower practice scores relative to that of students from high income class (OR 0.69 [95% CI: 0.54–0.89], p = 0.004; **Table 3**). In addition, we also found that students whose family members had been infected by COVID-19 had a more positive attitude (OR 1.53 [95% CI: 1.19–1.97], p = 0.001), while conversely students who were unaware of their history of contacts with COVID-19 patients or their history of COVID-19 infection had lower practice scores (OR 0.80 [95% CI: 0.68–0.93], p = 0.003, and OR 0.78 [95% CI: 0.66–0.92], p = 0.003, respectively)

**Table 1. Factors associated with knowledge on COVID-19 in the study population (n = 4870).**

| Variable | Univariate | | | Multivariate | | |
|---|---|---|---|---|---|---|
| | OR | 95% CI | P-value | OR | 95% CI | P-value |
| Age (years) | **1.17** | **1.13, 1.22** | **<0.001** | **1.06** | **1.00, 1.12** | **0.034** |
| Sex | | | | | | |
| Female | **1.19** | **1.04, 1.36** | **0.013** | **1.22** | **1.06, 1.40** | **0.006** |
| Male | ref | | | ref | | |
| Location | | | | | | |
| Sumatra | 0.86 | 0.64, 1.16 | 0.319 | **0.73** | **0.54, 0.99** | **0.044** |
| Central Indonesia[a] | 1.48 | 0.99, 2.21 | 0.057 | 1.41 | 0.93, 2.12 | 0.103 |
| Eastern Indonesia[b] | 0.88 | 0.67, 1.15 | 0.338 | **0.73** | **0.55, 0.97** | **0.028** |
| Java | ref | | | ref | | |
| Institution type | | | | | | |
| Public | **1.29** | **1.14, 1.46** | **<0.001** | **1.32** | **1.15, 1.50** | **<0.001** |
| Private | ref | | | ref | | |
| Academic level | | | | | | |
| Clinical | **2.06** | **1.78, 2.39** | **<0.001** | **1.66** | **1.35, 2.04** | **<0.001** |
| Pre-clinical | ref | | | ref | | |
| Living with[c] | | | | | | |
| Alone | 1.12 | 0.94, 1.34 | 0.202 | | | |
| Non-family | 1.08 | 0.62, 1.89 | 0.795 | | | |
| Family | ref | | | | | |
| Number of housemate (people)[c] | 1.01 | 0.99, 1.03 | 0.313 | | | |
| Living with children | | | | | | |
| Yes | 0.95 | 0.84, 1.07 | 0.401 | | | |
| No | ref | | | | | |
| Living with elderly | | | | | | |
| Yes | 1.14 | 0.99, 1.32 | 0.070 | 1.14 | 0.98, 1.32 | 0.084 |
| No | ref | | | ref | | |
| Marital status | | | | | | |
| Married | 0.94 | 0.37, 2.43 | 0.901 | | | |
| Divorced | 2.35 | 0.15, 37.67 | 0.545 | | | |
| Not married | ref | | | | | |
| Family income | | | | | | |
| ≤ IDR 1,500,000 | 1.01 | 0.76, 1.33 | 0.975 | | | |
| IDR 1,500,001–2,500,000 | 0.84 | 0.64, 1.11 | 0.229 | | | |
| IDR 2,500,001–3,500,000 | 0.92 | 0.75, 1.12 | 0.414 | | | |
| > IDR 3,500,000 | ref | | | | | |
| History of chronic illness | | | | | | |
| Yes | **1.40** | **1.10, 1.77** | **0.006** | **1.33** | **1.04, 1.70** | **0.022** |
| No | ref | | | ref | | |
| Volunteered in health sectors | | | | | | |
| Yes | **1.60** | **1.37, 1.86** | **<0.001** | **1.26** | **1.07, 1.49** | **0.007** |
| No | ref | | | ref | | |
| Volunteered in non-health sectors | | | | | | |
| Yes | **1.43** | **1.25, 1.63** | **<0.001** | **1.33** | **1.15, 1.52** | **<0.001** |
| No | ref | | | ref | | |
| Family members diagnosed with COVID-19 | | | | | | |
| Don't know | **0.83** | **0.69, 1.00** | **0.047** | 0.87 | 0.71, 1.07 | 0.180 |

*(Continued)*

**Table 1.** (Continued)

| Variable | Univariate | | | Multivariate | | |
|---|---|---|---|---|---|---|
| | OR | 95% CI | P-value | OR | 95% CI | P-value |
| Yes | 1.16 | 0.93, 1.45 | 0.184 | 1.05 | 0.82, 1.34 | 0.698 |
| No | ref | | | ref | | |
| Contacts with COVID-19 patients | | | | | | |
| Don't know | 0.88 | 0.75, 1.02 | 0.083 | 0.91 | 0.77, 1.07 | 0.254 |
| Yes | **1.68** | **1.23, 2.31** | **0.001** | 1.36 | 0.96, 1.93 | 0.086 |
| No | ref | | | ref | | |
| Had been a COVID-19 patient | | | | | | |
| Don't know | 0.89 | 0.75, 1.05 | 0.152 | 0.89 | 0.74, 1.06 | 0.192 |
| Yes[d] | 1.15 | 0.56, 2.39 | 0.669 | 0.85 | 0.39, 1.86 | 0.691 |
| No | ref | | | ref | | |

Texts in bold indicate statistical significance.

[a]Includes Sulawesi and Kalimantan.

[b]Includes Bali, Nusa Tenggara, Maluku, and Papua.

[c]Defined as the people living with the respondents at the time of questionnaire completion.

[d]Includes both confirmed and unconfirmed (suspected or probable) cases. CI, confidence interval; COVID-19, coronavirus disease 2019; IDR, Indonesian Rupiah; OR, odds ratio.

## Levels of trust toward COVID-19 health information sources

In addition to the knowledge, attitude, and practice toward COVID-19, we also explored the study population's levels of trust toward COVID-19-related national health information sources. We discovered that most of the participants considered that health information released by health institutions (89.3%), health experts (78.0%), and the government (70.8%) to be reliable. In contrast, only 26.0% and 12.7% of the participants deemed online news and media social to be trustworthy **(Fig 2)**. Nearly half of the participants also trusted the health information broadcasted by television and newspaper outlets (45.8% and 44.3%, respectively). No robust correlations could be established between trust in specific health information sources and the students' knowledge, attitude, or practice. Although the negative association between trust in social media and the students' knowledge was statistically significant, the magnitude of the correlation was clinically negligible ($\rho$ = -0.040; p = 0.005; **S5 Table**).

## Discussion

Medical education primarily aims to inculcate medical students' high-quality skills and competence in preparing them as future healthcare professionals. In a global health crisis, medical students play a pivotal role in raising public awareness, which in turn may contribute to successful emergency management by mitigating risks, supporting preventive measures, and minimizing negative psychological burdens [18]. To maximize the potentials of these students in educating the communities, it is thus important to explore their knowledge, attitude, and practice toward COVID-19. The present cross-sectional study showed that a majority of Indonesian medical students had a positive attitude and practice against COVID-19. However, this number was not accompanied by a proportionate number of students with adequate knowledge, indicating the urgent need to take active measures to keep these students updated with COVID-19-related evidence.

In the present study, female medical students were superior to males in terms of knowledge and practice. This finding is further validated by a meta-analysis demonstrating that women

**Table 2. Factors associated with attitude toward COVID-19 in the study population (n = 4870).**

| Variable | Univariate | | | Multivariate | | |
|---|---|---|---|---|---|---|
| | OR | 95% CI | P-value | OR | 95% CI | P-value |
| Age (years) | **1.05** | **1.01, 1.09** | **0.018** | 1.04 | 0.98, 1.09 | 0.196 |
| Sex | | | | | | |
| Female | 0.96 | 0.84, 1.09 | 0.492 | | | |
| Male | ref | | | | | |
| Location | | | | | | |
| Sumatra | 0.99 | 0.76, 1.31 | 0.967 | 0.92 | 0.69, 1.22 | 0.547 |
| Central Indonesia[a] | 1.00 | 0.66, 1.50 | 0.990 | 0.93 | 0.62, 1.42 | 0.749 |
| Eastern Indonesia[b] | 0.84 | 0.66, 1.07 | 0.151 | 0.83 | 0.64, 1.07 | 0.144 |
| Java | ref | | | ref | | |
| Institution type | | | | | | |
| Public | 1.11 | 0.99, 1.25 | 0.086 | 1.12 | 0.98, 1.23 | 0.093 |
| Private | ref | | | ref | | |
| Academic level | | | | | | |
| Clinical | 1.16 | 1.00, 1.35 | 0.050 | 1.00 | 0.81, 1.23 | 0.979 |
| Pre-clinical | ref | | | ref | | |
| Living with[c] | | | | | | |
| Alone | 0.92 | 0.78, 1.09 | 0.338 | | | |
| Non-family | 1.02 | 0.59, 1.75 | 0.953 | | | |
| Family | ref | | | | | |
| Number of housemate (people)[c] | 1.02 | 0.99, 1.04 | 0.177 | 1.02 | 0.99, 1.04 | 0.190 |
| Living with children | | | | | | |
| Yes | 1.04 | 0.92, 1.16 | 0.577 | | | |
| No | ref | | | | | |
| Living with elderly | | | | | | |
| Yes | 1.06 | 0.92, 1.22 | 0.415 | | | |
| No | ref | | | | | |
| Marital status | | | | | | |
| Married | 1.36 | 0.53, 3.50 | 0.530 | | | |
| Divorced | NE | NE | 0.999 | | | |
| Not married | ref | | | | | |
| Family income | | | | | | |
| $\leq$ IDR 1,500,000 | 1.01 | 0.77, 1.33 | 0.930 | 1.03 | 0.77, 1.36 | 0.865 |
| IDR 1,500,001–2,500,000 | **0.68** | **0.53, 0.87** | **0.002** | **0.70** | **0.54, 0.89** | **0.005** |
| IDR 2,500,001–3,500,000 | 0.89 | 0.73, 1.07 | 0.207 | 0.91 | 0.75, 1.10 | 0.328 |
| > IDR 3,500,000 | ref | | | ref | | |
| History of chronic illness | | | | | | |
| Yes | 1.05 | 0.82, 1.33 | 0.719 | | | |
| No | ref | | | | | |
| Volunteered in health sectors | | | | | | |
| Yes | **1.34** | **1.14, 1.57** | **<0.001** | **1.19** | **1.00, 1.41** | **0.045** |
| No | ref | | | ref | | |
| Volunteered in non-health sectors | | | | | | |
| Yes | **1.28** | **1.14, 1.45** | **<0.001** | **1.25** | **1.10, 1.41** | **0.001** |
| No | ref | | | ref | | |
| Family members diagnosed with COVID-19 | | | | | | |
| Don't know | 1.01 | 0.85, 1.20 | 0.909 | 1.00 | 0.83, 1.20 | 0.974 |

*(Continued)*

**Table 2.** (Continued)

| Variable | Univariate | | | Multivariate | | |
|---|---|---|---|---|---|---|
| | OR | 95% CI | P-value | OR | 95% CI | P-value |
| Yes | **1.63** | **1.29, 2.07** | **<0.001** | **1.53** | **1.19, 1.97** | **0.001** |
| No | ref | | | ref | | |
| Contacts with COVID-19 patients | | | | | | |
| Don't know | 1.03 | 0.89, 1.19 | 0.674 | 0.97 | 0.83, 1.14 | 0.726 |
| Yes | **1.65** | **1.15, 2.35** | **0.006** | 1.34 | 0.91, 1.98 | 0.137 |
| No | ref | | | ref | | |
| Had been a COVID-19 patient | | | | | | |
| Don't know | 1.15 | 0.99, 1.35 | 0.077 | 1.13 | 0.95, 1.34 | 0.174 |
| Yes[d] | 0.97 | 0.48, 1.98 | 0.933 | 0.60 | 0.28, 1.27 | 0.182 |
| No | ref | | | ref | | |

Texts in bold indicate statistical significance.

[a]Includes Sulawesi and Kalimantan.

[b]Includes Bali, Nusa Tenggara, Maluku, and Papua.

[c]Defined as the people living with the respondents at the time of questionnaire completion.

[d]Includes both confirmed and unconfirmed (suspected or probable) cases. CI, confidence interval; COVID-19, coronavirus disease 2019; IDR, Indonesian Rupiah; NE, not estimable; OR, odds ratio.

were 49.5% more likely to practice and adopt health-protective behaviors in the context of a pandemic outbreak [19]. Our study also revealed that a higher percentage of students from public medical schools demonstrated adequate knowledge towards COVID-19, while medical students from private institutions reported a higher level of practice. Although these results may have noted the importance of embedding and promoting equality between public and private medical institutions as well as between fellow public or private medical institutions themselves in terms of the medical curriculum adopted and opportunities to practice clinical skills, these findings should be interpreted cautiously, especially considering the complex interrelationship between the explored variables.

One of the most apparent findings to emerge from the analysis was that medical students in clinical years yielded higher knowledge scores compared to pre-clinical students. This result may be explained by the fact that medical students in their final years have been exposed to more experiences and learning opportunities in clinical setup, thereby offering a vast wealth of potential to help and contribute to the pandemic response. With the rigorous years of clinical training they have undergone, qualified clinical-year medical students may therefore contribute beyond being public educators by volunteering to clinically assist the healthcare workers in fighting against COVID-19 [20].

Our analysis further validated that voluntary participation, whether in health or non-health sectors, played a major role in determining one's levels of knowledge, attitude, and practice. In response to the ongoing COVID-19 pandemic, a plethora of volunteering efforts has been established and launched. Willingness to volunteer has been demonstrated to be higher in those who establish preparedness behavior and exhibit higher awareness of responsibility [13]. In addition, depending on the role and scope, volunteering might offer the opportunities to develop relevant knowledge and skills, a positive sense of community, and pro-social behavior [21]. These might explain the higher scores observed in the present survey among medical students with volunteering experience. Accordingly, based on our findings, incremental efforts should more specifically be made by the medical institutions in promoting volunteerism and

**Table 3. Factors associated with practice toward COVID-19 in the study population (n = 4870).**

| Variable | Univariate | | | Multivariate | | |
|---|---|---|---|---|---|---|
| | OR | 95% CI | P-value | OR | 95% CI | P-value |
| Age (years) | **1.04** | **1.00, 1.08** | **0.036** | 1.01 | 0.96, 1.07 | 0.626 |
| Sex | | | | | | |
| Female | **1.47** | **1.30, 1.67** | **<0.001** | **1.46** | **1.28, 1.65** | **<0.001** |
| Male | ref | | | ref | | |
| Location | | | | | | |
| Sumatra | 1.07 | 0.82, 1.39 | 0.631 | | | |
| Central Indonesia[a] | 0.79 | 0.53, 1.17 | 0.235 | | | |
| Eastern Indonesia[b] | 0.97 | 0.76, 1.22 | 0.772 | | | |
| Java | ref | | | | | |
| Institution type | | | | | | |
| Public | **0.72** | **0.64, 0.81** | **<0.001** | **0.71** | **0.63, 0.80** | **<0.001** |
| Private | ref | | | ref | | |
| Academic level | | | | | | |
| Clinical | 1.15 | 0.99, 1.32 | 0.060 | 1.07 | 0.88, 1.31 | 0.479 |
| Pre-clinical | ref | | | ref | | |
| Living with[c] | | | | | | |
| Alone | 1.08 | 0.91, 1.27 | 0.386 | | | |
| Non-family | 0.77 | 0.46, 1.30 | 0.323 | | | |
| Family | ref | | | | | |
| Number of housemate (people)[c] | 1.01 | 0.99, 1.03 | 0.329 | | | |
| Living with children | | | | | | |
| Yes | 1.01 | 0.90, 1.13 | 0.870 | | | |
| No | ref | | | | | |
| Living with elderly | | | | | | |
| Yes | 0.99 | 0.86, 1.12 | 0.825 | | | |
| No | ref | | | | | |
| Marital status | | | | | | |
| Married | 1.89 | 0.76, 4.68 | 0.172 | 1.83 | 0.73, 4.62 | 0.200 |
| Divorced | 0.94 | 0.06, 15.08 | 0.967 | 0.90 | 0.06, 14.65 | 0.941 |
| Not married | ref | | | ref | | |
| Family income | | | | | | |
| ≤ IDR 1,500,000 | 0.99 | 0.76, 1.29 | 0.946 | 1.00 | 0.76, 1.30 | 0.973 |
| IDR 1,500,001–2,500,000 | **0.71** | **0.56, 0.91** | **0.007** | **0.69** | **0.54, 0.89** | **0.004** |
| IDR 2,500,001–3,500,000 | 0.98 | 0.82, 1.17 | 0.797 | 0.99 | 0.82, 1.19 | 0.888 |
| > IDR 3,500,000 | ref | | | ref | | |
| History of chronic illness | | | | | | |
| Yes | 0.94 | 0.75, 1.18 | 0.571 | | | |
| No | ref | | | | | |
| Volunteered in health sectors | | | | | | |
| Yes | **1.30** | **1.12, 1.50** | **0.001** | **1.25** | **1.07, 1.47** | **0.006** |
| No | ref | | | ref | | |
| Volunteered in non-health sectors | | | | | | |
| Yes | **1.21** | **1.08, 1.36** | **0.001** | **1.16** | **1.03, 1.31** | **0.019** |
| No | ref | | | ref | | |
| Family members diagnosed with COVID-19 | | | | | | |
| Don't know | **0.76** | **0.65, 0.90** | **0.002** | 0.90 | 0.75, 1.07 | 0.236 |

*(Continued)*

**Table 3.** (Continued)

| Variable | Univariate | | | Multivariate | | |
|---|---|---|---|---|---|---|
| | OR | 95% CI | P-value | OR | 95% CI | P-value |
| Yes | 1.08 | 0.88, 1.34 | 0.457 | 1.08 | 0.86, 1.36 | 0.493 |
| No | ref | | | ref | | |
| Contacts with COVID-19 patients | | | | | | |
| Don't know | **0.71** | **0.62, 0.82** | **<0.001** | **0.80** | **0.68, 0.93** | **0.003** |
| Yes | 1.35 | 0.98, 1.85 | 0.065 | 1.26 | 0.89, 1.78 | 0.200 |
| No | ref | | | ref | | |
| Had been a COVID-19 patient | | | | | | |
| Don't know | **0.67** | **0.58, 0.78** | **<0.001** | **0.78** | **0.66, 0.92** | **0.003** |
| Yes[d] | 1.76 | 0.85, 3.64 | 0.126 | 1.54 | 0.72, 3.30 | 0.263 |
| No | ref | | | ref | | |

Texts in bold indicate statistical significance.

[a]Includes Sulawesi and Kalimantan.

[b]Includes Bali, Nusa Tenggara, Maluku, and Papua.

[c]Defined as the people living with the respondents at the time of questionnaire completion.

[d]Includes both confirmed and unconfirmed (suspected or probable) cases. CI, confidence interval; COVID-19, coronavirus disease 2019; IDR, Indonesian Rupiah; OR, odds ratio.

encouraging more medical students to partake in volunteering opportunities to gain indispensable learning opportunities and collaborate with other healthcare professionals [22].

Additionally, it was also evident from our study that poor practices were demonstrated among respondents who were unsure whether or not they had been in contact with any COVID-19 positive patient or had been infected with COVID-19. This finding might indicate that the lack of awareness surrounding COVID-19 negatively affects one's level practice. Therefore, a comprehensive approach to increase the awareness of surrounding environment of these students, and in a broader scope–the general population, is urgently required. This may be achieved through rigorous contact tracing, intensive risk communication, as well as mass education efforts [23].

With the exponential technological advancement over the past few decades, social media seems to be the most plausible way to promote public health behavioral change to increase COVID-19 protective measures [24]. However, our findings suggested that trust in social media as a source of COVID-19 health information was inversely associated with favorable practice in the study population. Although the correlation was clinically negligible, this result noted the importance of managing the information flow of social media, as a source of unfiltered and potentially misleading information, while simultaneously protecting one's freedom of speech. This might be attained by developing algorithms and capabilities to detect fake news, cultivating a standard of conduct in dealing with fake news, and increasing the media literacy and ethical standards of digital users [25]. Nevertheless, it is worth noting that further studies investigating the association between social media and COVID-19 preventive practice among the Indonesian general population are required to confirm our premises as the current study population was confined to medical students. Furthermore, the fact that virtually no other health information sources was significantly associated with the levels of knowledge, attitude, or practice toward COVID-19 in the study population may also indicate that some more influential resources such as scientific articles or medical textbooks, which were unexplored in this study, may impact the overall trend.

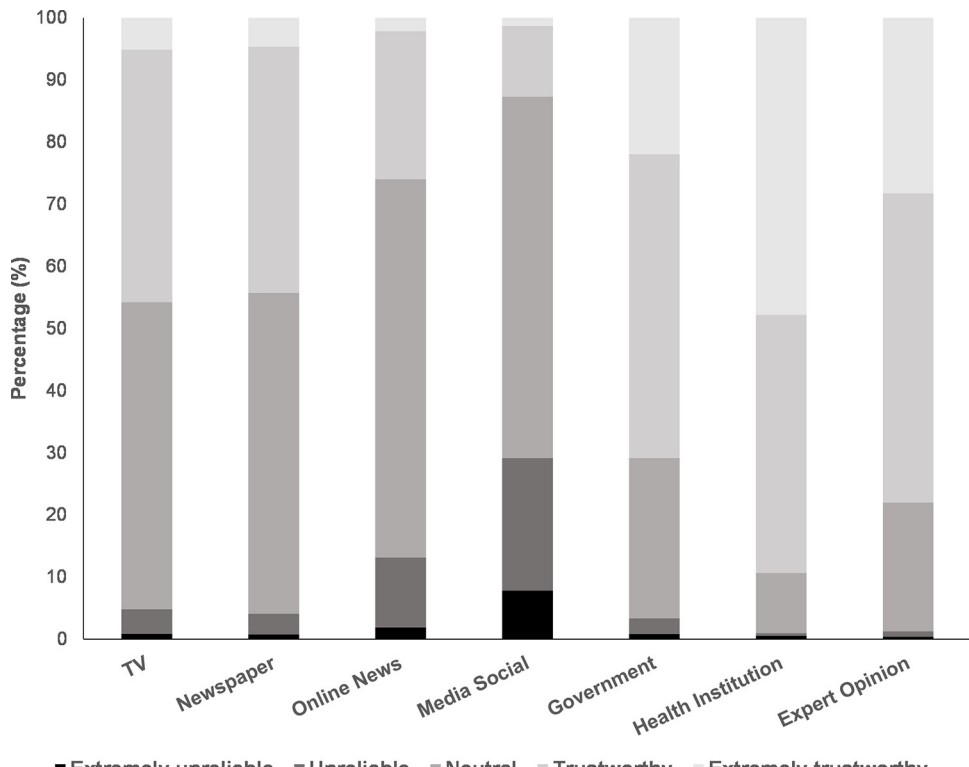

**Fig 2. Levels of trust of the study population toward COVID-19 health information released by various media outlet.**

Overall, the gaps in knowledge relating to COVID-19 persisted although most respondents in the study demonstrated a high level of attitude and a fair level of practice. These results showed that positive attitude and appropriate practices regarding COVID-19 documented among medical students might suggest their valuable role as role models for the general population. However, as future health professionals, demonstration of high standards of attitude and practice has to be aptly supported by excellence in clinical knowledge and understanding, especially if they are to be involved in global health emergencies. Addressing this knowledge gap will warrant not only a more effective public education but also a safer and more efficient involvement of medical students in public health emergencies beyond their role in raising public awareness [7].

One's level of knowledge is substantially influenced by an effective and efficient education system that plays a key role in ensuring high-quality teaching and learning. The low level of knowledge reported might be justified by the rapidly evolving COVID-19 evidence [26], which represents an enormous challenge to medical education and thus may subsequently hinder the delivery of educational materials. Our findings necessitate prompt actions by medical institutions to enhance the breadth of knowledge and understanding of Indonesian medical students in regards to COVID-19, particularly with reference to infection prevention and control principles. This was demonstrated by Boodman et al., who described the involvement of medical students in Canada to produce a weekly evidence-based newsletter designed to answer COVID-19 clinical questions raised by doctors. Besides improvements observed in research and inter-professional communication skills among the students, this strategy allowed medical students to gain a deeper understanding of COVID-19 while contributing in a concrete way to

the pandemic [27]. In addition, attention should also be paid to gradually allow medical students to engage safely in patient-based training with an appropriate balance of online and in-person learning. In-person activities can be conducted by mitigating the risk of physical contact with patients through physical distancing and suitable personal protective equipment [28].

The findings of our study might be of assistance and applicable for stakeholders and policymakers in designing and transforming existing public health interventions and medical curriculum to equip medical students with the appropriate tools to adapt during a global health crisis. The current pandemic has corroborated the noteworthiness of implementing exhaustive and systematic disaster training programs as part of the medical school curriculum to fight not only the current pandemic but also future unforeseeable global health crises. A key priority should therefore be to plan these dedicated programs to strengthen students' disaster and pandemic preparedness against similar global health calamities [29].

This study has several strengths and limitations. The relatively large sample size and the wide geographical reach contributed to the strength of the study. Furthermore, the questionnaire had previously been validated and yielded a fair reliability, thus further ascertaining the validity of our findings. However, the study was limited by the unbalanced distribution of pre-clinical and clinical medical students, which could have potentially limited the generalizability of the study results. The generalizability of our findings might also be affected by the constantly evolving evidence on and situation due to COVID-19, thereby implying that the knowledge, attitude, and practice of medical students in Indonesia found in this study may change over time. Moreover, due to the cross-sectional nature of the survey, we were not able to disentangle the directionality of the relationships observed. An additional uncontrolled factor was the possibility that possible confounding variables, which were not scrutinized in this study, might affect our results. To the best of our knowledge, this is the first reported study assessing the levels of knowledge, attitude, and practice toward the COVID-19 disease among Indonesian medical students, thus providing key parameters for policymakers and institutions in formulating effective strategies and tools to enhance the medical students' potentials in raising public awareness and protective practices in the current COVID-19 pandemic and prospective potential public health emergencies.

## Conclusion

Undergraduate medical students in Indonesia had a considerably positive attitude and practice against COVID-19. However, further interventions are required as these figures were not complemented with a proportionate number of students with adequate knowledge. Such interventions should aim to keep the students updated with COVID-19 evidence and simultaneously providing them with opportunities to contribute to the pandemic as public educators and role models for communities, while also equipping them with appropriate knowledge and skills to prepare for future public health emergencies. In turn, this approach may create a positive feedback loop enhancing the students' knowledge, attitude, and practice, which were positively intercorrelated in this study.

## Supporting information

**S1 Table. Characteristics of the study population.**
(DOCX)

**S2 Table. Item-specific responses on the participants' knowledge on COVID-19.**
(DOCX)

**S3 Table. Item-specific responses on the participants' attitude towards COVID-19.**
(DOCX)

**S4 Table. Item-specific responses on the participants' practice towards COVID-19.**
(DOCX)

**S5 Table. Correlation between the students' level of trust in health information sources with knowledge, attitude, and practice toward COVID-19.**
(DOCX)

## Acknowledgments

The authors would like to thank the collaborators of the MEDICO-19 Research Group for their contributions in data collection: Alanis Maryjane Mamahit, An Nahl Aulia Hakim, Astrid Cynthia Latief, Ervin Widyantoro Pramono, Feliani Sanjaya, Felix Kurniawan Adithia, Gabriel Tandecxi, Melisa Canggra, Muhammad Mufaiduddin, Rowaida Putri Anggaily Bian, Yoriska, and Zefo Kiyosi Wibowo. The authors would also like to express their gratitude to Prof. Mohamed Izham Mohamed Ibrahim of the College of Pharmacy, QU Health, Qatar University for the insights in the development of the questionnaire, as well as Mrs. Bira Arnetha and Ms. Sandhita Utami for the support in administering the project. Lastly, the authors would like to thank fellow students and every institutional official for the help in disseminating the questionnaire.

## Author Contributions

**Conceptualization:** Indah Suci Widyahening, Gilbert Lazarus, Azis Muhammad Putera, Nico Gamalliel.

**Data curation:** Imam Adli, Gilbert Lazarus.

**Formal analysis:** Imam Adli, Gilbert Lazarus.

**Funding acquisition:** Indah Suci Widyahening, Gilbert Lazarus, Lyanna Azzahra Baihaqi, Ardi Findyartini.

**Investigation:** Gilbert Lazarus, Jason Phowira, Bagas Ariffandi, Azis Muhammad Putera, David Nugraha, Nico Gamalliel.

**Methodology:** Indah Suci Widyahening, Gilbert Lazarus, Ardi Findyartini.

**Project administration:** Gilbert Lazarus, Lyanna Azzahra Baihaqi, Ardi Findyartini.

**Supervision:** Indah Suci Widyahening, Ardi Findyartini.

**Validation:** Indah Suci Widyahening, Ardi Findyartini.

**Visualization:** Gilbert Lazarus.

**Writing – original draft:** Imam Adli, Gilbert Lazarus, Jason Phowira, Lyanna Azzahra Baihaqi, Bagas Ariffandi.

**Writing – review & editing:** Imam Adli, Indah Suci Widyahening, Gilbert Lazarus, Jason Phowira, Lyanna Azzahra Baihaqi, Azis Muhammad Putera, David Nugraha, Nico Gamalliel, Ardi Findyartini.

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
