## [Decision Letter · Decision Letter 0]

26 Oct 2021

PONE-D-21-12734Knowledge, attitude, and practice related to the COVID-19 pandemic among undergraduate medical students in Indonesia: a nationwide cross-sectional studyPLOS ONE

Dear Dr. Indah Suci Widyahening,

Thank you for submitting your manuscript to PLOS ONE. After careful consideration, we feel that it has merit but does not fully meet PLOS ONE’s publication criteria as it currently stands. Therefore, we invite you to submit a revised version of the manuscript that addresses the points raised during the review process.

We look forward to receiving your revised manuscript.

Kind regards,

Muhammad Junaid Farrukh

Academic Editor

PLOS ONE

Journal Requirements:

2. Please provide additional details regarding participant consent. In the ethics statement in the Methods and online submission information, please ensure that you have specified (1) whether consent was suitably informed and (2) what type you obtained (for instance, written or verbal). If your study included minors under age 18, state whether you obtained consent from parents or guardians. If the need for consent was waived by the ethics committee, please include this information.

Reviewers' comments:

Reviewer's Responses to Questions

**Comments to the Author**

1. Is the manuscript technically sound, and do the data support the conclusions?

Reviewer #2: Partly

Reviewer #3: Partly

2. Has the statistical analysis been performed appropriately and rigorously? 

Reviewer #2: Yes

Reviewer #3: No

3. Have the authors made all data underlying the findings in their manuscript fully available?

Reviewer #2: Yes

Reviewer #3: Yes

4. Is the manuscript presented in an intelligible fashion and written in standard English?

Reviewer #2: Yes

Reviewer #3: Yes

5. Review Comments to the Author

Reviewer #2: Dear Authors,

Please consider my following suggestions to further improve the manuscript:

1. To state any pilot study conducted prior to actual data collection.

2. To state any reminder mechanism to increase the number of participants

3. To state any mechanism to avoid duplicates of response (i.e. from same respondent)

4. To report the overall response rate

5. To state whether or not the data was collected during the lockdown period in Indonesia.

6. To state any limitation of the study, particularly related to No. 5 above. This is important as there are always new information and development of CoVID-19 between now and back in end of 2019.

Thank you and all the best.

Reviewer #3: 33: This study aims to assess . . . .

36: Socio-demographics

39: Specify the 64.9% and 51.5% positive attitude and practice

98: Measurement tools

101: GL, IA, and AMP write full abbreviation

104 and 106: Socio-demographic

122: Explain what do you mean by two 12 five-point Likert, In table it is 5 Likert scale?

166: What do you mean by older students here? Female and older students. If possible, specify which age group, how do you interpretate older students. Median age is 20 IQR 19-21.

168: Table 1 is representing the “Odd Ratio (OR)” and need to rephrase the interpretation in the text, how it has been interpretated female and older students were more knowledge able

228: correlation value and p value are different from table S5

Specify the cut off score knowledge, Attitude, and Practice score in Table S1

Do mention percentage and frequency n (%) in table S3 and S4

Kindly check the divorced percentage and frequency in Table 2

Concern with your ref value in table 2 and 3. Some independent variables ref is first variable and in some it is last and some in between… be consistent

6. PLOS authors have the option to publish the peer review history of their article (what does this mean?). If published, this will include your full peer review and any attached files.

Reviewer #2: No

Reviewer #3: No

---

## [Author Response · Author response to Decision Letter 0]

14 Dec 2021

Dear Editor-in-chief and respected reviewers,

Thank you for the comments from the editor and reviewers on our manuscript. We really appreciate the constructive and detailed feedback on our manuscript. We have revised the current submitted manuscript based on the reviewers’ feedback as detailed in the "Response to reviewer (see appendix page 71-75)". The texts highlighted in yellow refer to changes made following the reviewers’ suggestions.

We hope you will kindly reconsider our submission and we are looking forward to the publication of our manuscript.

Thank you very much in advance for your kind consideration.

Sincerely yours,

Representing all authors

Indah Suci Widyahening, MD, M.S., M.Sc-CMFM, PhD

Department of Community Medicine, 

Faculty of Medicine Universitas Indonesia, Jakarta 10430, Indonesia

Phone/Fax No: +62 21 3141066

Email Address: indah_widyahening@ui.ac.id

---

## [Editor Report · Decision Letter 1]

6 Jan 2022

Knowledge, attitude, and practice related to the COVID-19 pandemic among undergraduate medical students in Indonesia: a nationwide cross-sectional study

PONE-D-21-12734R1

Dear Indah Suci Widyahening

We’re pleased to inform you that your manuscript has been judged scientifically suitable for publication and will be formally accepted for publication once it meets all outstanding technical requirements.

Kind regards,

Muhammad Junaid Farrukh

Academic Editor

PLOS ONE

Additional Editor Comments (optional):

dear author, after reviewing your revised manuscript, we are happy to inform you that your manuscript is now suitable for publication.
---

## [Editor Report · Acceptance letter]

11 Jan 2022

PONE-D-21-12734R1 

Knowledge, attitude, and practice related to the COVID-19 pandemic among undergraduate medical students in Indonesia: a nationwide cross-sectional study 

Dear Dr. Widyahening:

I'm pleased to inform you that your manuscript has been deemed suitable for publication in PLOS ONE. Congratulations! Your manuscript is now with our production department. 

Kind regards, 

on behalf of

Dr. Muhammad Junaid Farrukh 

Academic Editor

PLOS ONE